# Political endorsement by *Nature* and trust in scientific expertise during COVID-19

**Floyd Jiuyun Zhang** ⬤ ✉

High-profile political endorsements by scientific publications have become common in recent years, raising concerns about backlash against the endorsing organizations and scientific expertise. In a preregistered large-sample controlled experiment, I randomly assigned participants to receive information about the endorsement of Joe Biden by the scientific journal *Nature* during the COVID-19 pandemic. The endorsement message caused large reductions in stated trust in *Nature* among Trump supporters. This distrust lowered the demand for COVID-related information provided by *Nature*, as evidenced by substantially reduced requests for *Nature* articles on vaccine efficacy when offered. The endorsement also reduced Trump supporters' trust in scientists in general. The estimated effects on Biden supporters' trust in *Nature* and scientists were positive, small and mostly statistically insignificant. I found little evidence that the endorsement changed views about Biden and Trump. These results suggest that political endorsement by scientific journals can undermine and polarize public confidence in the endorsing journals and the scientific community.

Scientific organizations and publications have become increasingly involved in electoral politics. In the run-up to the 2020 US presidential election, numerous influential scientific publications, including *Nature, Scientific American*, the *Lancet*, the *New England Journal of Medicine* and *Science*, published editorial pieces criticizing then-president Donald Trump's mishandling of the COVID-19 pandemic and his antagonistic attitudes towards science. Most of these journals urged voters to replace Trump. Among them, *Nature, Scientific American* and the *Lancet* explicitly endorsed his challenger Joe Biden[1]. This marked the first time *Scientific American* or the *Lancet* had made a political endorsement. These publications were joined by 81 American Nobel laureates in endorsing Biden's candidacy[2].

The increased political engagement by scientists raises concerns that their endorsements cause right-wing backlash[3]. Trust in the scientific community has been declining in the United States for decades, with the most pronounced decline among those on the political right[4]. During the COVID-19 pandemic, such scepticism towards scientific expertise reduced compliance with public health interventions[5] and may explain the partisan difference in compliance[6,7], with important implications for public health outcomes[8]. By endorsing a Democratic candidate in a polarizing presidential election during the pandemic,

scientists risk intensifying existing distrust from a large segment of the population, particularly because these endorsements were widely reported by conservative media outlets[9,10].

The possibility of a right-wing backlash is consistent with the literature on affective political polarization in the United States[11], which often finds that associating individuals or entities with a political party increases out-party animosity towards them. It is also consistent with Bayesian models of information economics and decision theory, which predict that an agent uncertain about the quality of an information source may judge its quality by the degree to which its messages conform to the agent's prior[12–14]. However, it would not be unreasonable to expect no backlash. Given the prestige of publications such as *Nature*, it is unclear whether their credibility would be judged on the basis of a political endorsement—in Bayesian terms, there may or may not be enough prior uncertainty with respect to source quality to trigger a substantively meaningful update. In addition, expressed partisan hostilities are sometimes "cheerleading"[15] that does not translate into behaviours when there are stakes such as health risks. Finally, research shows that priming Americans about COVID-19 reduces affective polarization[16]. Whether these endorsements have any effects on trust and behaviours is thus an empirical question.

Graduate School of Business, Stanford University, Stanford, CA, USA. ✉e-mail: floydjz@stanford.edu

This paper presents findings from a preregistered online experiment examining the effects of *Nature*'s 2020 endorsement of Joe Biden for US president amid the COVID-19 pandemic. Conducted in late July and early August 2021, the experiment randomly assigned participants to receive information about *Nature*'s endorsement, while the control group received irrelevant information.

In addition to examining the consequences of well-intentioned political activism in the scientific community, this paper contributes to the literature on scientific communication, trust in scientific expertise[4,17–20], political endorsement[21–24], non-political consequences of political polarization in the United States[6,25–29], and the social and behavioural aspects of COVID-19 responses[30]. This paper presents an experimental study of the effects that scientists' political activities have on trust in scientists. For an observational study on public opinion effects of the 2017 March for Science rallies, see Motta[19]. Also closely related is an experimental study by Kotcher et al.[31], which examined policy advocacy instead of political advocacy and found that climate-related policy advocacy has limited or no effect on the perceived credibility of the communicating scientists and the scientific community. In addition, though political endorsements have been extensively studied, the effect of endorsements on the endorser or public perceptions thereof remains understudied. This study fills this gap.

## Results

### The experiment

The experiment took the form of an online survey with randomized components. At the beginning of the survey, the participants were screened for attention and asked about their political beliefs. The experimental sample consists of 4,260 individuals and is broadly representative of the US adult population along most demographic dimensions (Table 1). However, the sample is skewed towards Biden supporters, as indicated by responses to a (pretreatment) question asking the participants their preference between Biden and Trump: 55.14% of the participants preferred Biden, while 35.06% favoured Trump.

To examine the effect of *Nature*'s endorsement, I randomly assigned half of the participants to read a short message summarizing *Nature*'s endorsement piece for Biden (left panel of Fig. 1). The message highlighted *Nature*'s criticism of Trump's mishandling of the COVID-19 pandemic and its expectation that Biden would do better. To make sure the message was credible, the summary was followed by a screenshot of the endorsement piece's title, lead paragraph and cover picture from *Nature*'s official website, as well as a link to the piece. Finally, the participants were reminded that *Nature* is "one of the most-cited and most prestigious peer-reviewed scientific journals in the world".

The control participants were assigned to read a message about *Nature*'s announcement of its new visual designs for its website and print copies, instead of the endorsement (right panel of Fig. 1). The message was presented in the same format as the endorsement message that the treatment participants saw. The text was also followed by a screenshot and a link to *Nature*'s announcement of the new design, as well as the same reminder of *Nature*'s scientific prominence. Nowhere in the survey were the control participants informed of the endorsement.

After the treatment or control message, the participants were shown batteries of questions and messages eliciting measures of (1) trust in *Nature*, (2) assessments of Biden and Trump, (3) the demand for COVID-19 information provided by *Nature*, (4) the persuasiveness of a climate-change-related message attributed to *Nature*, and (5) trust in scientists in general.

Analyses were conducted on participants who indicated some degree of support for either Biden or Trump in the aforementioned pretreatment candidate preference question. This excluded a small minority (8.80%) of participants who stated that they supported "someone else", as their political predispositions were unknown and difficult to interpret[32].

**Table 1 | Sample breakdown by demographics**

| | Sample (%) | US adult population (%) |
|---|---|---|
| **Gender** | | |
| Female | 53.0% | 51.3% |
| Male | 46.5% | 48.7% |
| **Age** | | |
| <18 | 0.2% | |
| 18–24 | 9.7% | 11.7% |
| 25–34 | 19.2% | 17.9% |
| 35–44 | 19.0% | 16.6% |
| 45–54 | 13.6% | 15.7% |
| 55–65 | 20.8% | 18.0% |
| >65 | 17.4% | 20.14% |
| **Hispanic** | | |
| Hispanic/Latino/Spanish | 16.1% | 16.6% |
| **Race and ethnicity (survey category)** | | |
| Black/African American | 11.2% | |
| White | 77.8% | |
| American Indian/Alaska Native | 1.3% | |
| Asian | 5.0% | |
| Native Hawaiian/Pacific Islander | 0.5% | |
| Other | 4.2% | |
| **Race and ethnicity (ACS general category)** | | |
| Black/African American | | 11.8% |
| White | | 65.1% |
| American Indian/Alaska Native | | 0.9% |
| Chinese | | 1.5% |
| Japanese | | 0.3% |
| Asian or Pacific Islander | | 4.3% |
| Other race | | 6.3% |
| Two major races | | 9.3% |
| Three or more major races | | 0.6% |
| **Highest level of education** | | |
| No high school | 2.4% | 10.7% |
| High school graduate/GED | 23.3% | 26.8% |
| Began college, no degree | 20.1% | 21.6% |
| Associate's/technical degree | 13.9% | 8.4% |
| Bachelor's degree | 26.9% | 20.26% |
| Postgraduate/professional degree | 13.5% | 12.3% |
| **Census region** | | |
| Northeast | 19.7% | 17.3% |
| Midwest | 19.6% | 20.7% |
| South | 39.1% | 38.2% |
| West | 21.7% | 23.8% |

Sample *N* = 4,260. Only participants who passed the attention check are included in the sample percentages. The US adult population estimates are based on the 2020 American Community Survey (ACS), extracted from IPUMS USA[53]. My survey and the ACS use slightly different race categories that are not directly comparable, so they are summarized in separate columns.

### Stated trust in *Nature*

I first examined the effect of seeing the endorsement on stated trust in *Nature*. The survey elicited the participants' confidence in the journal

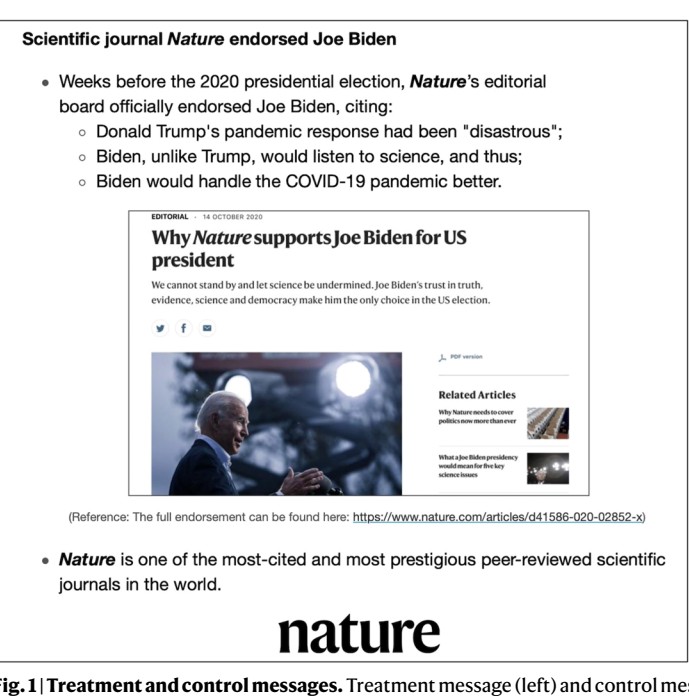

**Fig. 1 | Treatment and control messages.** Treatment message (left) and control message (right). Credits for left panel: main, Roberto Schmidt/AFP/Getty; top thumbnail, Willy Kurniawan/Reuters; bottom thumbnail, Alex Wong/Getty.

along two dimensions—namely, its informedness and impartiality. These outcomes were captured by two questions asking the participants to report how much they trusted *Nature* for (1) being informed when providing advice on science-related issues facing society and (2) giving their unbiased opinions to the public, to the best of their knowledge, when contentious issues are concerned. The questions and the distributions of responses by political alignment and treatment status are presented in Figs. 2 and 3.

Table 2 reports the results from regression analyses of the treatment effects. The responses to the two questions are mapped on five-point scales and, as are all other outcomes analysed in Table 2, standardized as *z* scores with mean 0 and standard deviation 1. The endorsement had large negative effects on Trump supporters' trust in *Nature*'s informedness ($t(3,881) = -16.49$; $P < 0.001$; $\beta = -0.854$; 95% confidence interval (CI), $(-0.955, -0.752)$) and impartiality ($t(3,881) = -12.71$; $P < 0.001$; $\beta = -0.633$; 95% CI, $(-0.730, -0.534)$). The effects on Biden supporters are positive but small, significant only for the 'informed' outcome ($t(3,881) = 3.50$; $P < 0.001$; $\beta = 0.108$; 95% CI, $(0.047, 0.169)$). Taken together, the endorsement appears to have further polarized trust in *Nature*.

The estimates suggest that the polarizing effect of the endorsement is greater than the baseline difference between Biden supporters and Trump supporters. When untreated, Trump supporters' confidence in *Nature*'s informedness and impartiality is 0.387 ($t(3,881) = -10.03$; $P < 0.001$; $\beta = -0.387$; 95% CI, $(0.577, 0.734)$) and 0.655 ($t(3,881) = -16.40$; $P < 0.001$; $\beta = -0.655$; 95%, CI, $(0.311, 0.463)$) standard deviations lower than that of Biden supporters', respectively. The treatment pulls them apart by an additional 0.962 standard deviations ($t(3,881) = -15.96$; $P < 0.001$; $\beta = -0.96$; 95% CI, $(0.844, 1.080)$) and 0.678 standard deviations ($t(3,881) = -11.53$; $P < 0.001$; $\beta = -0.678$; 95% CI, $(0.562, 0.793)$), respectively. Depending on the measure looked at, the treatment increases the trust gap between the two categories by a factor of 2 or 3.5.

## Heterogeneity by prior beliefs

The effects on trust in *Nature* might be explained by two possible theoretical mechanisms: information and context. The informational explanation posits that the treatment provides new information to Bayesian agents, who then update their beliefs about *Nature*. In contrast, contextual explanations (for example, priming) suggest that the treatment condition may create a context in which *Nature*'s political activism is especially salient and thereby leads to a (potentially short-lived) effect on sentiments towards the journal.

To distinguish the two explanations empirically, I leveraged their different predictions with respect to the treatment effect heterogeneity by prior beliefs about the endorsement. If the informational mechanism is at work, the effect of the message should be greater for individuals who did not expect *Nature* to make political endorsements ex ante, as the treatment induces in them larger updates of beliefs. Contextual explanations such as priming would not predict such heterogeneity, since the contextual difference between the treatment and the control conditions is the same regardless of the participants' prior knowledge or expectation. To test for such heterogeneity, I elicited the participants' prior beliefs by asking how likely they thought it was that *Nature* had made any political endorsement in the 2020 presidential election, before showing the treatment or control message. Figure 4 presents the estimated effects on trust in *Nature* for each prior belief level.

For Trump supporters, not expecting the endorsement is clearly associated with a larger decrease in trust when told that *Nature* did endorse Biden. The treatment effect for Trump supporters who did not expect *Nature* to endorse at all ('not likely at all') is two to three times as large as that for Trump supporters who fully expected it ('they definitely did'). If prior belief is incorporated as a one-dimensional five-point scale, along with the treatment indicator and their interaction, in a linear regression model that explains Trump supporters' trust in *Nature*, the interaction between prior belief and treatment is significant for both trust in informedness ($t(1,528) = -3.09$; $P = 0.002$; $\beta = -0.159$; 95% CI, $(-0.261, -0.058)$) and trust in impartiality ($t(1,528) = -2.24$; $P = 0.025$; $\beta = -0.112$; 95% CI, $(-0.212, -0.014)$). Among Biden supporters, the size of the estimated increase in trust in *Nature*'s knowledge also seems larger for those who thought endorsement was unlikely, but the interaction is statistically insignificant. There is no discernible pattern in effects on Biden supporters' trust in *Nature*'s impartiality, which is unsurprising given that the aggregate effect on this outcome is not significant for them.

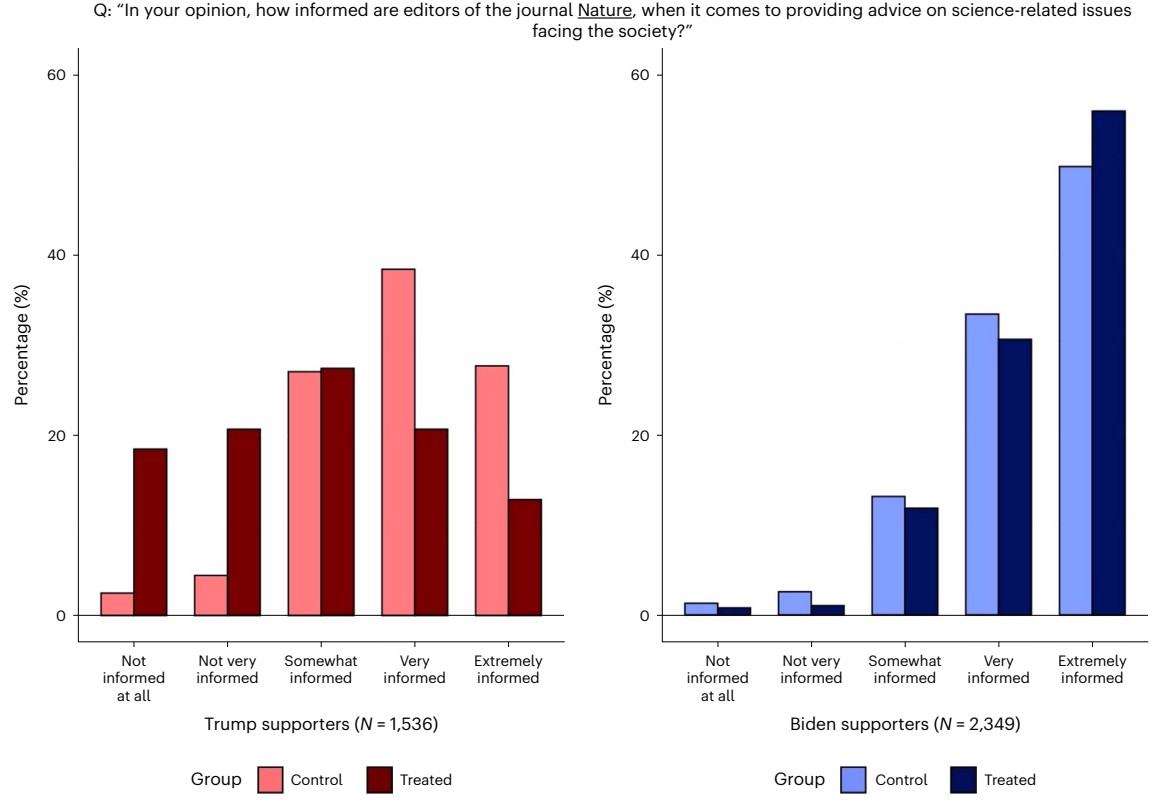

Q: "In your opinion, how informed are editors of the journal <u>Nature</u>, when it comes to providing advice on science-related issues facing the society?"

**Fig. 2 | Trust in *Nature*'s knowledge.** Stated trust in *Nature* being informed.

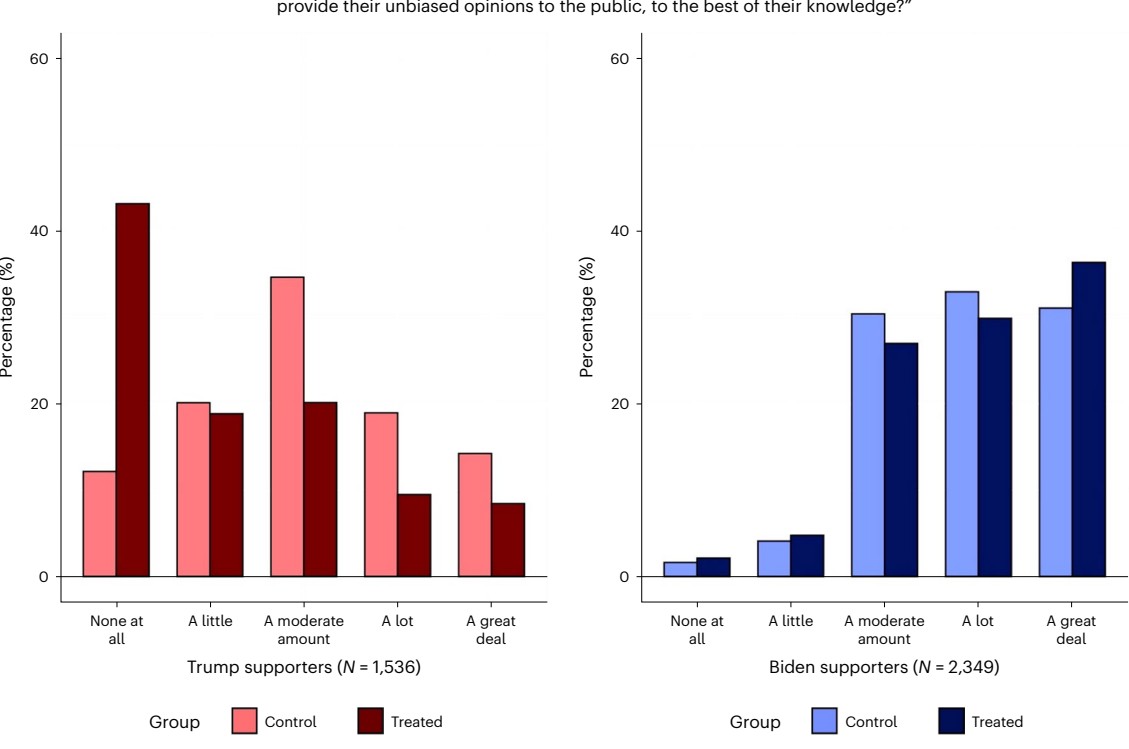

Q: "When contentious or divisive issues are concerned, how much confidence do you have in the editorial board of <u>Nature</u> to provide their unbiased opinions to the public, to the best of their knowledge?"

**Fig. 3 | Trust in *Nature*'s impartiality.** Stated trust in *Nature* being unbiased.

## Endorsement persuasiveness

I looked into whether the endorsement was successful at changing people's minds about Biden's and Trump's relative competence by focusing on the issues that the endorsement highlights—namely, pandemic response and attitudes towards science. The next three rows of Table 2 report the findings. There is little evidence that the endorsement persuaded participants. In addition to the general absence of statistical significance, the upper bounds of the 95% CIs never exceed 0.12

## Table 2 | Experiment results

| Outcome z score | $\widehat{\text{CATE}}$ | | Baseline difference |
| | Trump supporters | Biden supporters | |
| | (Robust s.e.) | (Robust s.e.) | (Robust s.e.) |
|---|---|---|---|
| Nature is informed when providing advice | −0.854 | 0.108 | −0.387 |
| | (0.0518) | (0.0309) | (0.0386) |
| | P=0.000 | P=0.000 | P=0.000 |
| Nature is unbiased on contentious issues | −0.633 | 0.0450 | −0.655 |
| | (0.0498) | (0.0313) | (0.0399) |
| | P=0.000 | P=0.151 | P=0.000 |
| Biden would have handled COVID better than Trump had he been president in 2020 | 0.0556 | 0.0448 | −1.642 |
| | (0.0322) | (0.0238) | (0.0282) |
| | P=0.085 | P=0.061 | P=0.000 |
| Trump would have handled COVID better than Biden if he were still president now (2021) | −0.0241 | −0.0168 | 1.628 |
| | (0.0329) | (0.0240) | (0.0287) |
| | P=0.465 | P=0.485 | P=0.000 |
| Biden is better at making use of scientific knowledge for decision-making than Trump | 0.0449 | 0.0135 | −1.775 |
| | (0.0354) | (0.0140) | (0.0267) |
| | P=0.204 | P=0.335 | P=0.000 |
| Participant requests Nature's article for information about vaccine efficacy against new variants | −0.285 | 0.048 | −0.386 |
| | (0.0463) | (0.0409) | (0.0455) |
| | P=0.000 | P=0.238 | P=0.000 |
| US scientists are informed when providing advice | −0.130 | 0.0485 | −0.756 |
| | (0.0532) | (0.0335) | (0.0442) |
| | P=0.015 | P=0.148 | P=0.000 |
| US scientists are unbiased on contentious issues | −0.161 | 0.0161 | −0.937 |
| | (0.0516) | (0.0310) | (0.0422) |
| | P=0.002 | P=0.604 | P=0.000 |
| ≥90% of climate scientists agree that human-caused climate change is real | −0.0461 | 0.0265 | −0.590 |
| | (0.0558) | (0.0350) | (0.0463) |
| | P=0.409 | P=0.449 | P=0.000 |
| Human-caused climate change is real | −0.0232 | 0.0147 | −0.974 |
| | (0.0582) | (0.0266) | (0.0456) |
| | P=0.690 | P=0.582 | P=0.000 |

N=3,885. CATE, conditional average treatment effect. $\widehat{\text{CATE}}$ for Trump (Biden) supporters is the estimated treatment effect for Trump (Biden) supporters. 'Baseline difference' is the mean difference between Trump supporters and Biden supporters within the control group. The sample includes 1,173 control Biden supporters, 1,176 treatment Biden supporters, 766 control Trump supporters and 770 treatment Trump supporters. The outcomes are z scores with mean 0 and standard deviation 1. All estimates are from ordinary least squares estimation of a linear regression model, described in the Methods. Robust standard errors are given in parentheses. All null hypotheses testing was done with two-sided t-tests.

standard deviations and are typically much smaller. The null results for the three outcomes are also supported by Bayes factors of 49.755, 940.039 and 394.801, respectively, from the corresponding Bayesian linear regression models.

The construction of these questions is fairly conducive to finding persuasive effects, as COVID-19 and science are the focuses of the endorsement message. The null findings also distinguish my results from 'backfire effects' in which opposing information intensifies individuals' pre-existing opinions on the subject matter communicated[33,34].

### Demand for information

Do the shifts in stated trust in Nature translate into changes in behaviours such as demand for COVID-related information provided by Nature? To study this question, I look at how the participants chose to acquire information from a menu of sources. I prompted the participants with a message reminding them of emerging COVID variants and

encouraging them to "stay informed about vaccine efficacy against new COVID variants". The topic, which was salient and affected everyone's health and safety, was chosen to increase the stake of the choice. As the situation was rapidly changing, most people were unlikely to have kept track of it and likely to want to learn more.

The message offered links to "easy-to-read" articles about "new variants and how well available vaccines perform against them" from a variety of sources. Specifically, the participants could choose to read from Nature, the Mayo Clinic, unspecified "news media websites" or any combination of the three sources, or not to read about the topic at all.

The endorsement led to a statistically significant 14.2-percentage-point reduction in the frequency at which Trump supporters requested Nature articles ($t(3,881) = -6.15$; $P < 0.001$; $\beta = -0.142$; 95% CI, (−18.750, −9.688))—a 38.3% decline relative to control Trump supporters, who requested Nature articles 37.1% of the time. Biden supporters, in contrast, selected Nature 56.4% of the time under the control

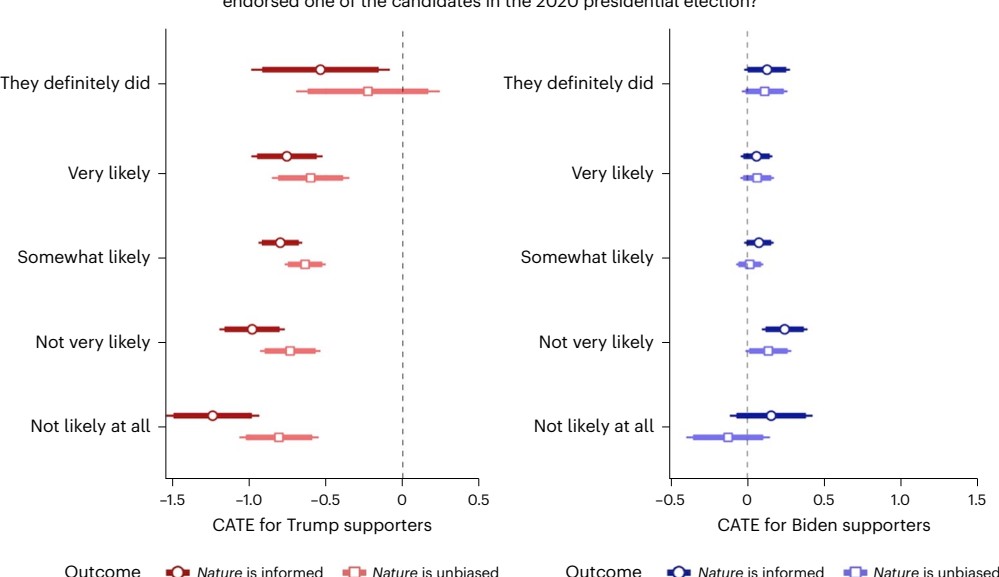

**Fig. 4 | Treatment effect heterogeneity by prior belief.** The estimates are presented as CATE point estimates ± 95% (thin bars) and 90% CIs (thick bars). $N$ = 3,885 experiment participants.

condition. The estimated treatment effect for Biden supporters is positive but small and statistically insignificant ($t(3,881)$ = 1.18; $P$ = 0.238; $\beta$ = 0.048; 95% CI, (−0.032, 0.128)). The $z$-score versions of these results are presented in Table 2.

It is worth noting that the experiment was conducted early during the Delta-variant-induced surge of US cases in summer 2021, when the variant first became salient in Americans' consciousness. Google Trends[35] shows that US search frequencies for the keywords 'Delta' and 'variant' were at their all-time peak when the experiment was conducted (28 July–10 August). The search frequency for 'vaccine' also reached a local peak. Given the thirst for information on this topic at the time, it is all the more telling that Trump supporters became substantially less likely to want to read about it from *Nature* after exposure to the endorsement information.

**Trust in scientists in general**

Next, I considered possible reputational externalities on the scientific community. That is, the endorsement may have affected trust not just in *Nature* but also in scientists generally. The next set of outcomes are based on a pair of questions that parallel the two questions about trust in *Nature*'s knowledge and impartiality, with '*Nature*' replaced by 'US scientists'.

Figures 5 and 6 show the distribution of responses by group. Under the control condition, Trump supporters were less likely to report high levels of trust towards US scientists than Biden supporters, and the gap is larger for treated participants. The regression analyses reported in Table 2 show statistically significant negative treatment effects for Trump supporters (informedness: $t(3,881)$ = −2.44; $P$ = 0.015; $\beta$ = −0.130; 95% CI, (−0.234, −0.026); unbiasedness: $t(3,881)$ = −3.12; $P$ = 0.002; $\beta$ = −0.161; 95% CI, (−0.262, −0.060)), albeit considerably smaller than the corresponding effects on trust in *Nature*. The estimates for Biden supporters are positive but small and statistically insignificant (informedness: $t(3,881)$ = 1.45; $P$ = 0.148; $\beta$ = 0.048; 95% CI, (−0.017, 0.114); unbiasedness: $t(3,881)$ = 0.52; $P$ = 0.604; $\beta$ = 0.016; 95% CI, (−0.045, 0.077)).

**Climate change**

I also examined whether the endorsement affected *Nature*'s credibility when it communicates scientific consensus on other domains.

The questionnaire displayed a quote from an editorial piece by *Nature Climate Change*, stating that 97% of climate scientists agree that climate change is real and caused by human activities. (On the questionnaire, the quote was mistakenly attributed to *Nature* instead of *Nature Climate Change*, due to my misunderstanding of the distinction at the time.) The treatment group were again reminded that *Nature* endorsed Biden. The participants were then asked whether they believed the statement about climate scientists' consensus and whether they believed in human-caused climate change. The signs of all estimates are consistent with the results for trust in *Nature*, but the effects are small and insignificant (Table 2).

**Discussion**

This study shows that electoral endorsements by *Nature* and potentially other scientific journals or organizations can undermine public trust in the endorser, particularly among supporters of the out-party candidate. This has negative impacts on trust in the scientific community as a whole and on information acquisition behaviours with respect to critical public health issues. Positive effects among supporters of the endorsed candidate are null or small, and they do not offset the negative effects among the opposite camp. This probably results in a lower overall level of public confidence and more polarization along the party line. There is little evidence that seeing the endorsement message changes opinions about the candidates.

These results indicate that seeing the endorsement substantially reduced stated trust in *Nature*'s informedness and impartiality among Trump supporters. Treated Trump and Biden supporters became two to four times more polarized than the control participants on these stated measures of trust in *Nature*. The effect was greater for participants who did not expect *Nature* to make an endorsement ex ante. The endorsement also significantly dampened Trump supporters' demand for COVID-related information provided by *Nature*. When prompted to acquire information about emerging COVID-19 variants and vaccine efficacy, treated Trump supporters were 38% less likely than control Trump supporters to request stories from *Nature*'s website, indicating that the decrease in stated trust has behavioural consequences. The endorsement also significantly reduced trust in scientists in general among Trump supporters, creating a reputational externality on the entire scientific community.

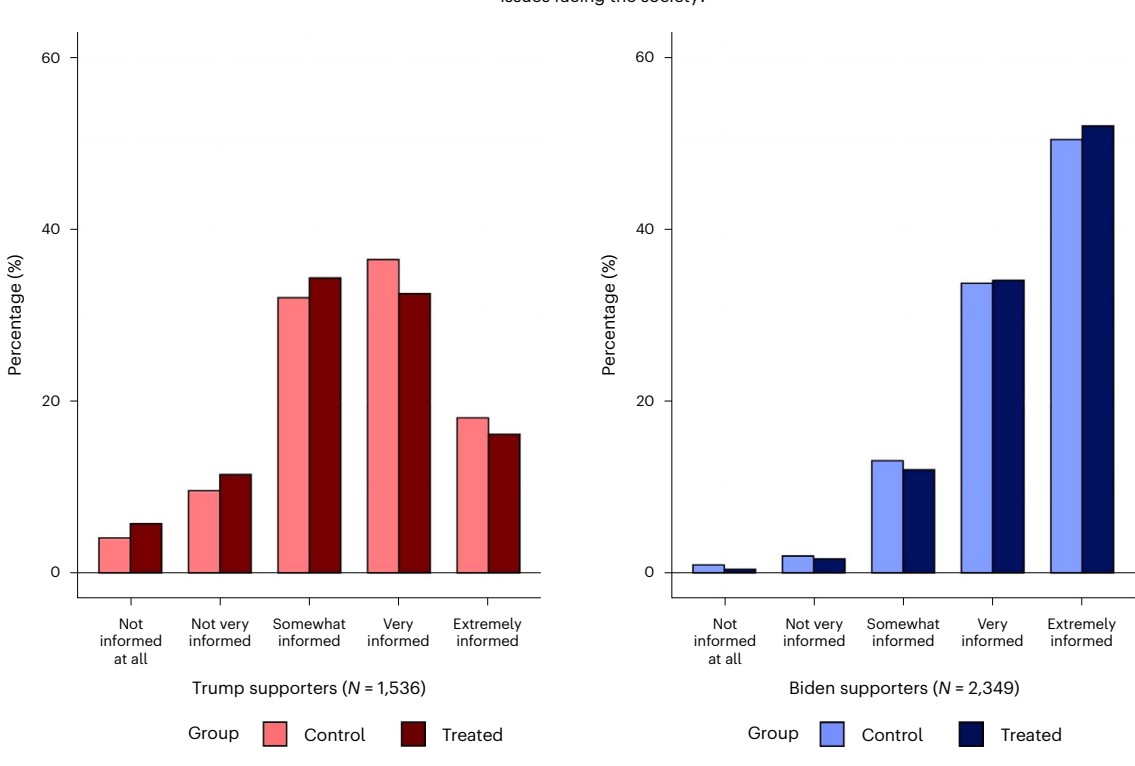

Q: "In your opinion, how informed are U.S. scientists, when it comes to providing advice on science-related issues facing the society?"

**Fig. 5 | Trust in scientists' knowledge.** Stated trust in US scientists being informed.

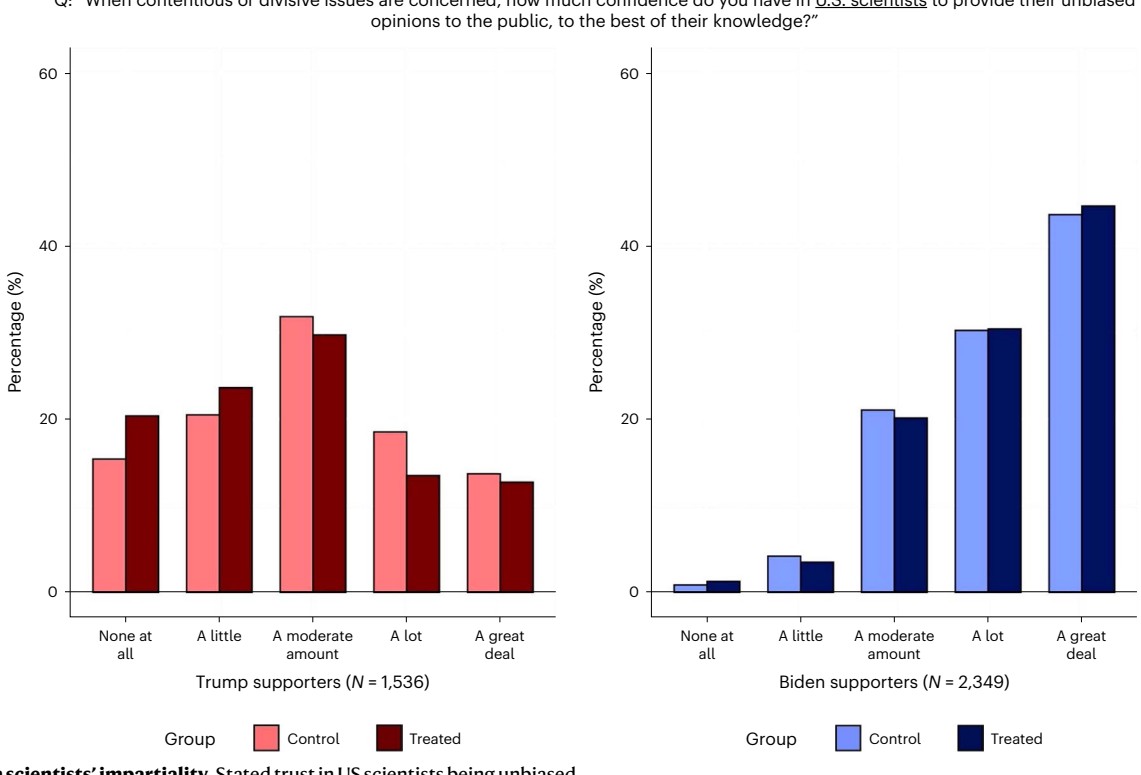

Q: "When contentious or divisive issues are concerned, how much confidence do you have in U.S. scientists to provide their unbiased opinions to the public, to the best of their knowledge?"

**Fig. 6 | Trust in scientists' impartiality.** Stated trust in US scientists being unbiased.

The point estimates for Biden supporters suggest that the endorsement may have slightly increased Biden supporters' confidence in *Nature* and scientists. However, the estimated effects are substantively small and often statistically insignificant.

Finally, there is little evidence that the endorsement changed participants' opinions about the two presidential candidates with respect to the issues that the endorsement piece highlights. The findings suggest that, if the objective is to shape public opinions and the political

environment in a way that is conducive to scientific endeavours and evidence-based policies, political endorsements given by scientific journals may have substantial downsides and little upside.

These results are not due to Biden's relative unpopularity since late 2021. The experiment was conducted in late July and early August 2021, when Biden's approval rating was above 50% and only slightly below its peak[36]. Furthermore, a pilot study I conducted found qualitatively similar effects in May 2021, when Biden's overall popularity was still at its 'honeymoon' level, with 71% of Americans, including 47% of Republicans, approving his handling of COVID[37].

It is instructive to interpret the results in the light of science communication theories. Donner[38] proposes conceptualizing advocacy in scientists' public communications as a continuum, where higher levels of advocacy are characterized by more normative judgements and greater influence of the scientist's worldview. He also theorizes that greater degrees of advocacy lead to greater "professional risks" such as "alienating those…[in the] audience with opposing political views." This phenomenon that messages can damage the persuasiveness of the communicators is referred to as source derogation in communication and psychology research[39]. The type of political endorsement studied here, which advocates voting as a tactic to bring about better policy actions, is close to the highest level of advocacy that Donner[38] considered, which is characterized by being motivated by questions such as 'Should certain tactics be employed to bring about specific actions (for example, participating in a protest against fossil fuel development or a pipeline)?' This study's findings thus strongly support Donner's hypothesis that a high level of advocacy leads to high risks to the communicator's perceived credibility. The difference between these findings and that of Kotcher et al.[31], that climate-related policy advocacy has limited or no effect on the scientist's credibility, also fits the theory, since supporting specific policies (actions) corresponds to a lower level of advocacy than advocating political tactics (voting in this case) to bring about these actions on Donner's continuum. This study can thus be seen as complementary to Kotcher et al.[31]. In addition, this study's finding of effect heterogeneity by the audience's political predisposition is consistent with research on motivated reasoning—that is, the tendency for agents to systematically reject information that contradicts their own deeply held beliefs[40,41].

The findings have implications beyond this specific context. Education has been playing an increasingly important role in political alignment across Western democracies over the past several decades[42]. In such settings, scientists and science sceptics are probably, more than ever, represented by opposing sides of the partisan divide. Dynamics similar to the one studied here may thus play out in other times and places[43,44].

This study also has several limitations and qualifications worth pointing out. First, the experiment demonstrates that *Nature*'s endorsement decision can cause large negative reputation effects, but it does not show that the endorsement did have such an impact or that other similar statements from scientific journals must produce such outcomes. In addition, the finding of sizable negative reputation externality on the scientific community as a whole may not be very widely generalizable beyond well-known top journals like *Nature*, which may be seen as an exemplar of the scientific community because of its reputation[45,46]. Finally, because this was a one-shot experiment, it remains an open question whether the effects are short- or long-lasting.

Future research should address some of the weaknesses of this study. Most importantly, techniques such as follow-up surveys should be employed to determine, as soon as possible, whether the attitudinal and behavioural effects found in this study are long-lived. Furthermore, future research should also examine whether an explicit endorsement is necessary to produce these effects—would the impacts be different if the editorial statement had made the same criticism of the Trump administration without explicitly endorsing an alternative? Finally, it would also be of interest to explore empirically whether these findings apply only to scientific journals or whether they are generalizable to other types of organizations.

## Methods

### Ethics, consent and preregistration

The study protocol was approved by the Stanford Research Compliance Office. The participants accepted the consent form and agreed to participate in the study in exchange for a cash payment agreed on between the participants and the panel research company. The consent form is included in the Supplementary Information along with the complete survey instrument.

The experiment was double-registered in the American Economics Association RCT Registry (https://www.socialscienceregistry.org/trials/7007) and in the Open Science Foundation Registry (https://osf.io/ge2m8/). The former was preregistered on 27 July 2021, before the experiment launched; the latter was preregistered on 2 August 2021, during the response collection period and before I first accessed or analysed any portion of the data. The contents of the two registrations do not have any substantive differences.

The only deviation from the preregistered protocol is the sample size. The preregistration states that 4,000 responses would be collected—that is, after attention screening but before excluding participants who were neither Biden supporters nor Trump supporters from the analyses. Although I explicitly requested only 4,000 responses from Lucid Theorem, by the time I downloaded and started analysing the data, 4,260 eligible responses had already been collected. All analyses presented are based on this larger-than-planned sample. This deviation is small in magnitude. Supplementary Method 3 presents the analyses using only the first eligible 4,000 responses collected, which yields virtually identical estimates.

Two additional analyses specified in the pre-analysis plan—namely, multiple testing corrections and lasso regression adjustments with auxiliary features—are presented in Supplementary Tables 2 and 3 to keep the main text concise, as the findings are not meaningfully different from what is in the main text.

### Sample

The experiment took place between 28 July and 10 August 2021 and was conducted using a national (US) sample of 4,260 individuals who had been screened for attention (Supplementary Note 1); the resulting group of participants demonstrated high attention and responded strongly to the treatment message (Supplementary Notes). The sample size calculation is based on data from a smaller-scale pilot experiment; the sample is also larger than those reported in important and influential publications using similar methods in related literature[30,31]. The participants were invited to share their opinions "in a survey about current events" and participated voluntarily by signing a consent document. The sample was recruited from Lucid, which helps researchers build samples representative of the US adult population along targeted demographic dimensions, with two caveats. First, the distributions of characteristics not targeted by Lucid Theorem may not reflect those of the population. For example, Democrats tend to be over-represented in Lucid Theorem samples. Second, the attention check that I placed, which screens out inattentive participants, could alter the distribution if attention is correlated with other participant characteristics.

In practice, however, the second caveat does not seem to pose a meaningful challenge to the representativeness of my sample. Table 1 shows post-attention-check sample breakdowns by key demographic and geographical categories, which broadly mirror those of the US adult population. However, the sample does not seem representative with respect to political leaning. Specifically, Biden supporters are probably over-represented, if the 2020 presidential election result is taken as the benchmark (Supplementary Table 1). It is worth pointing out that this skewed distribution is not the result of screening for attention but

rather is due to Democrats being regularly over-represented in Lucid Theorem samples.

## Survey instruments
A complete copy of the survey instruments used in the experiment is included in the Supplementary Information.

## Randomization and blinding
The treatment was randomized on the individual level and blocked on prior political position (Supplementary Table 1) to ensure finite sample balance. The randomization was implemented with Qualtrics randomizers. There are five blocks. One half of the participants in each block received the treatment.

The experiment is double-blind: the participants were not informed of their own treatment assignment, and the experiment administration was automated.

## Analysis
Throughout the analyses, I focused on heterogeneity by the participants' baseline political opinions, particularly by whether they supported Biden or Trump. This opinion was elicited by a pretreatment question about their "hypothetical vote intention" asking who they would vote for if they "were to choose again" between Biden and Trump (since the study took place many months after the election). The distribution of answers among participants who passed the attention check is reported in Supplementary Table 1. A majority 55.14% of the participants responded either "Definitely Biden" or "Probably Biden". I labelled them Biden supporters. The 35.06% who favoured Trump were labelled Trump supporters. In my analysis, I dropped the 8.80% who stated that they would vote for "someone else", since there is no clear interpretation for their pretreatment political alignment.

**Statistical information.** The experimental data were analysed by estimating the following linear regression model:

$$Y_i = \alpha + \beta D_i \times TS_i + \gamma D_i \times (1 - TS_i) + \delta TS_i + \epsilon_i \tag{1}$$

where $Y_i$ is the outcome for participant $i$, $D_i$ is a dummy variable indicating that participant $i$ is in the treatment group, $TS_i$ is a dummy variable indicating that $i$ is a Trump supporter and $\epsilon_i$ is a heteroskedastic error term.

This specification was chosen, among various equivalent models, for its ease of interpretation. In particular, since I dropped participants who were coded as neither Trump supporters nor Biden supporters, the omitted category is control Biden supporters. $\beta$ and $\gamma$ thus represent treatment effects for Trump supporters and for Biden supporters, respectively. $\delta$ is the baseline difference between the two categories. In Table 2, the estimates of $\beta$, $\gamma$ and $\delta$ are presented in columns 1, 2 and 3, respectively. The analyses reported in Table 2 and the main text do not use any covariates in addition to the variables included in equation (1).

The Bayes factors reported in the paper are derived from a Bayesian linear regression model with the same regression equation (equation (1)), normal coefficient prior, and normal error with the Jeffreys variance prior.

**Software.** The data were collected using Qualtrics XM; they were analysed and visualized using RStudio Version 1.4.1717 (ref. [47]), StataMP 14.0 (ref. [48]) and the crossEstimation R package[49].

## Supplementary analyses
**Controlling for covariates.** The survey collected a rich set of covariates capturing detailed characteristics of the participants. As specified in the pre-analysis plan, in addition to the parsimonious ordinary least squares estimation described in the previous section (which produces the results presented in the main text), I implemented regression adjustment estimation using lasso regression to improve the precision of the estimates[49,50]. The method combines unbiased estimates of the treatment effects with regularized control of auxiliary features. Presented in Supplementary Table 2, the regression adjustment estimates have slightly smaller standard errors but do not meaningfully alter the qualitative or quantitative results.

**Multiple testing corrections.** The main analyses involved testing multiple hypotheses. I performed multiple testing corrections using the sharpened false discovery rate[51,52] in accordance with the preregistered analysis plan. The adjusted $q$ values are presented in Supplementary Table 2, which do not change the conclusions in a substantively meaningful way.

## Reporting summary
Further information on research design is available in the Nature Portfolio Reporting Summary linked to this article.

## Data availability
The dataset analysed in the study is available in Harvard Dataverse as 'Replication data for: Political endorsement by *Nature* and trust in scientific expertise during COVID-19 (Zhang, 2023)', https://doi.org/10.7910/DVN/KBIRPN.

## Code availability
The code that reproduces the results is available in Harvard Dataverse as 'Replication data for: Political endorsement by *Nature* and trust in scientific expertise during COVID-19 (Zhang, 2023)', https://doi.org/10.7910/DVN/KBIRPN.

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

## Acknowledgements

I thank A. Acharya, M. Gentzkow and N. Malhotra for their continuous guidance; K. Casey, N. Halevy, S. Jha and K. Shotts for their comments and feedback; and B. Ginn and the Stanford GSB Behavioral Lab for help with subject recruitment. For funding support, I thank the Stanford Graduate School of Business's PhD Fellowship Program and the Stanford Center for American Democracy (SCAD) at the Institute for Research in the Social Sciences (IRiSS). The funder had no role in study design, data collection and analysis, decision to publish or preparation of the manuscript beyond the PhD research resources it provides.

## Competing interests

The author declares no competing interests.

## Additional information

**Correspondence and requests for materials** should be addressed to Floyd Jiuyun Zhang.

# Reporting Summary

## Statistics

For all statistical analyses, confirm that the following items are present in the figure legend, table legend, main text, or Methods section.

| n/a | Confirmed | |
|---|---|---|
| ☐ | ☒ | The exact sample size (*n*) for each experimental group/condition, given as a discrete number and unit of measurement |
| ☒ | ☐ | A statement on whether measurements were taken from distinct samples or whether the same sample was measured repeatedly |
| ☐ | ☒ | The statistical test(s) used AND whether they are one- or two-sided *Only common tests should be described solely by name; describe more complex techniques in the Methods section.* |
| ☐ | ☒ | A description of all covariates tested |
| ☐ | ☒ | A description of any assumptions or corrections, such as tests of normality and adjustment for multiple comparisons |
| ☐ | ☒ | A full description of the statistical parameters including central tendency (e.g. means) or other basic estimates (e.g. regression coefficient) AND variation (e.g. standard deviation) or associated estimates of uncertainty (e.g. confidence intervals) |
| ☐ | ☒ | For null hypothesis testing, the test statistic (e.g. *F*, *t*, *r*) with confidence intervals, effect sizes, degrees of freedom and *P* value noted *Give P values as exact values whenever suitable.* |
| ☒ | ☐ | For Bayesian analysis, information on the choice of priors and Markov chain Monte Carlo settings |
| ☒ | ☐ | For hierarchical and complex designs, identification of the appropriate level for tests and full reporting of outcomes |
| ☒ | ☐ | Estimates of effect sizes (e.g. Cohen's *d*, Pearson's *r*), indicating how they were calculated |

*Our web collection on statistics for biologists contains articles on many of the points above.*

## Software and code

Policy information about availability of computer code

| | |
|---|---|
| Data collection | I used Qualtrics XM to program and to distribute the experimental questionnaire, and to collect the resulting data. |
| Data analysis | I analyzed and visualized my data in Stata MP 14 and RStudio 1.4. In particular, the lasso regression adjustments are implemented using the crossEstimation package (https://github.com/swager/crossEstimation, the latest version as of Jan. 01 2023, last updated Feb. 20 2017) developed by Wager et al (2016). |

For manuscripts utilizing custom algorithms or software that are central to the research but not yet described in published literature, software must be made available to editors and reviewers. We strongly encourage code deposition in a community repository (e.g. GitHub). See the Nature Portfolio guidelines for submitting code & software for further information.

## Data

Policy information about availability of data

All manuscripts must include a data availability statement. This statement should provide the following information, where applicable:
- Accession codes, unique identifiers, or web links for publicly available datasets
- A description of any restrictions on data availability
- For clinical datasets or third party data, please ensure that the statement adheres to our policy

All data used in this study is generated specifically for the study during the experiment. The author will provide the dataset to editors and/or reviewers during the review process upon request. If accepted, the author will make the dataset publicly available on Harvard Dataverse before publication .

# Human research participants

Policy information about <u>studies involving human research participants and Sex and Gender in Research.</u>

| | |
|---|---|
| Reporting on sex and gender | I collected gender information by asking the following question in my survey questionnaire: "What's your gender?". The response options are "Female", "Male", and "Others". In my analysis sample (N=4,260), 2,257 (52.98%) identify as "Female"; 1,982 (46.53%) identify as "Male;"21 (0.49%) identify as "Others".  The only place this information is used is in Appendix A, where I control for demographics. There's no gender-based analysis, since the heterogeneity of interest is prior political opinions given the research questions. |
| Population characteristics | I collected information about participants' gender, age, race, education, state of residence, political opinions, etc., in the survey questionnaire. The distribution is presented in Tables 1 and 2 of the manuscript. The sample is largely representative of the U.S.  adult population, but Biden supporters seem to be over-represented relative to Trump supporters. |
| Recruitment | The respondents were recruited via Lucid Theorem, who aggregate respondents from various primary sources (online marketing panels. etc.) . Respondents were asked to share their opinions in "a survey about current events" in exchange for a monetary payment. The respondents signified consent by agreeing to a consent form and were screened for attention. These recruitment steps could introduce bias, but the resulting analysis sample is fairly represented of the U.S. adult population in terms of demographics (Table 1). The sample does not seem to be representative in terms of politics, with Biden supporters overrepresented. This does not introduce biased to the results since all estimates are conditional on prior political opinions. Given these, I think the estimates should be reasonably representative. |
| Ethics oversight | Stanford University IRB. (Protocol number: IRB-60462) |

Note that full information on the approval of the study protocol must also be provided in the manuscript.

# Field-specific reporting

Please select the one below that is the best fit for your research. If you are not sure, read the appropriate sections before making your selection.

☐ Life sciences  ☒ Behavioural & social sciences  ☐ Ecological, evolutionary & environmental sciences

For a reference copy of the document with all sections, see nature.com/documents/nr-reporting-summary-flat.pdf

# Behavioural & social sciences study design

All studies must disclose on these points even when the disclosure is negative.

| | |
|---|---|
| Study description | This is a quantitative experimental study with online participants. The experiment take the form of an online survey, with randomized information about Nature's endorsement of Joe Biden. Participants are then asked about their political and scientific attitudes. Statistical analyses are conducted to estimate the causal effect of seeing the endorsement on participants's views toward the journal Nature, U.S. scientists, Joe Biden, and Donald Trump. |
| Research sample | An online sample collected through Lucid Theorem that is broadly representative of the U.S. adult population in terms of age, sex, race, education level, and region of residence. The sample is chosen because of its demographic representativeness and ease of access. |
| Sampling strategy | Lucid Theorem constructs sample representative of the U.S. adult population from a pool of online respondents. The sample size is based on back-of-envelope calculation using data I collected in a pilot study. |
| Data collection | The respondents are given link to my Qualtrics questionnaire. Qualtrics records their responses. The experimental process is online and automated. No researcher is present when subjects go through the experiment. |
| Timing | July 28 - August 10, 2021 |
| Data exclusions | 375 completed responses (8.80% of the sample) are excluded because the subjects stated that they would vote for "someone else" (that is, neither Biden nor Trump). This is because their political positions don't have clear interpretation. I committed to this exclusion rule in my pre-registration /pre-analysis plan. See "Method" section in my manuscript for references to my pre-registration /pre-analysis plan. |
| Non-participation | 221 subjects who did not choose to agree to the consent document were screened out automatically prior to taking the survey. 1,710 of the consenting subjects are removed because they failed the attention check (See "Subject attention" subsection of the "Method" section). Of the 4,460 who consented and passed the attention check, exactly 200 did not complete the questionnaire. |
| Randomization | Randomization is implemented via Qualtrics randomizer. Before randomization, the survey asked each participant who would they vote for if they were to choose again between Biden and Trump. There are five response options to the question "Definitely Biden", |

"Probably Biden", "Definitely Trump", "Probably Trump", and someone else. The treatment is randomized within each of these five blocks to ensure finite sample balance. One half of the subjects in each block receive the treatment.

# Reporting for specific materials, systems and methods

We require information from authors about some types of materials, experimental systems and methods used in many studies. Here, indicate whether each material, system or method listed is relevant to your study. If you are not sure if a list item applies to your research, read the appropriate section before selecting a response.

## Materials & experimental systems

| n/a | Involved in the study |
|-----|----------------------|
| ⊠ ☐ | Antibodies |
| ⊠ ☐ | Eukaryotic cell lines |
| ⊠ ☐ | Palaeontology and archaeology |
| ⊠ ☐ | Animals and other organisms |
| ⊠ ☐ | Clinical data |
| ⊠ ☐ | Dual use research of concern |

## Methods

| n/a | Involved in the study |
|-----|----------------------|
| ⊠ ☐ | ChIP-seq |
| ⊠ ☐ | Flow cytometry |
| ⊠ ☐ | MRI-based neuroimaging |

