## [Peer Review File · Nature Human Behaviour]

Peer Review Information

Journal: Nature Human Behaviour

Manuscript Title: Political Endorsement by *Nature* and Trust in Scientific Expertise During COVID-19

Corresponding author name(s): Floyd Jiuyun Zhang

Reviewer Comments & Decisions:

Decision Letter, initial version:
--

25th August 2022

Dear Mr Zhang,

Thank you once again for your manuscript, entitled "Political Endorsement by Scientific Organizations Reduce and Polarize Trust in Scientific Expertise During COVID-19," and for your patience during the peer review process.

Your manuscript has now been evaluated by 2 reviewers, whose comments are included at the end of this letter. Although the reviewers find your work to be of interest, they also raise some important concerns. We are very interested in the possibility of publishing your study in *Nature Human Behaviour*, but would like to consider your response to these concerns in the form of a revised manuscript before we make a decision on publication.

Specifically, we think that it is important to address the reviewer concerns about the framing of the results. Please revise your title and manuscript to be specific about the questions that your research can and cannot answer. In addition, we ask you to follow the reviewer advice and elaborate on potential limitations of your findings as well as implications for future research.

In sum, we invite you to revise your manuscript taking into account all reviewer and editor comments. We are committed to providing a fair and constructive peer-review process. Do not hesitate to contact us if there are specific requests from the reviewers that you believe are technically impossible or

unlikely to yield a meaningful outcome.

We hope to receive your revised manuscript within two months. I would be grateful if you could contact us as soon as possible if you foresee difficulties with meeting this target resubmission date.

- Include a "Response to the editors and reviewers" document detailing, point-by-point, how you addressed each editor and referee comment. If no action was taken to address a point, you must provide a compelling argument. When formatting this document, please respond to each reviewer comment individually, including the full text of the reviewer comment verbatim followed by your response to the individual point. This response will be used by the editors to evaluate your revision and sent back to the reviewers along with the revised manuscript.
- Highlight all changes made to your manuscript or provide us with a version that tracks changes.

[REDACTED]

We look forward to seeing the revised manuscript and thank you for the opportunity to review your work. Please do not hesitate to contact me if you have any questions or would like to discuss these revisions further.

Sincerely,

Samantha Antusch

Samantha Antusch, PhD
Editor
Nature Human Behaviour

Reviewer expertise:

Reviewer #1: advocacy in science/trust in science ; survey experiments ; science communication

Reviewer #2: political science ; science communication ; endorsements ; persuasion

REVIEWER COMMENTS:

Reviewer #1:
Remarks to the Author:

This manuscript reports the findings of a pre-registered, randomized, controlled experiment that tested the immediate attitudinal and behavioral impacts of exposure to information about Nature's endorsement of Biden for President in 2020. I find the research to be well-conceived, appropriately conducted, and nicely written (in plain language, which is always of plus for research with important societal implications). The findings of the study are dramatic, and potentially very important. I recommend that it be accepted for publication with only minor revisions.

Major issues:

The conclusion of the paper—“(t)his study shows that electoral endorsements by scientific organizations reduce public trust in the endorser, particularly among supporters of the out-party candidate”—is overstated in my view. The study clearly shows that this CAN happen and that the effect can potentially be quite large, but it does not show that it DID happen in the 2020 U.S. Presidential election (only in a lab study conducted after the fact), or that it WILL happen in other instances in the future. In other words, I encourage the author to tone down the conclusion and make clear what was actually proven (versus the broader set of concerns that these important findings pose).

A related point is that the paper currently does not speak to either limitations of the study, or to priorities for future research. Given the importance of the findings, I feel strongly both of these features must be included.

The title of the paper, and the body of the paper, refers to Nature as a “scientific organization.” It is more accurate to call Nature a “scientific journal” and to limit claims to scientific journals rather than scientific organizations. The boomerang effects identified may—indeed, is likely to—also occur when scientific organizations (e.g., AAAS) make political endorsements, but that has not yet been tested or proven.

The paper addresses relevant theory only superficially in the introduction (although, oddly, it includes an extensive summary of the findings—which takes up approximately half of the introduction section), and not at all in the discussion. I suspect the superficial reference to theory in the introduction was in interest of moving readers quickly into the research itself—which is fine—but the paper will be stronger if some consideration of theory is incorporated into the discussion section.

The author speculates that the impacts of learning about the endorsement are likely to be long-lived, but frankly, I found the author's argument to be hard to follow; it should be clarified. More importantly, it should be clearly acknowledged that the impacts may NOT be long-lived, and that additional research should be conducted, ASAP, to determine the duration of impacts.

While the research was pre-registered, no hypotheses were stated in the pre-registration materials, nor in the paper itself. It isn't clear to me how readers should interpret pre-registered research questions (rather than pre-registered hypotheses). I would like to see the author address that issue in the discussion.

Minor issues:

Kotcher et al (2017) tested the impact of advocacy statements on trust in the communicating scientist and in the scientific community more broadly; they found limited to no impact. That research is sufficiently relevant to warrant consideration in the current paper. [Reference: Kotcher et al. (2017) Does engagement in advocacy hurt the credibility of scientists? Results from a randomized national survey experiment. Environmental Communication.]

Table 1 could be strengthened in the following ways:

- Drop the N column (as it is largely redundant with the % column)
- Add a column to show % from the current census data (to compare the sample to the overall population)
- I suggest limiting the numbers to one place after the decimal (e.g., 46.5%)

On page 5, the verbs in the 2nd paragraph are in the present tense (e.g., are) rather than in the past tense (were). Given that the actions occurred in the past, I think referring to them as past is appropriate.

There is inconsistency in how one of the "trust in Nature" variables is referred to in the paper. In some places it is referred to as "informed" and in other places "competent."

There is a misspelling in the first sentence of the 2nd paragraph of the discussion: "results".

Reviewer #2:

Remarks to the Author:

Review of "Political Endorsement by Scientific Organizations Reduce and Polarize Trust in Scientific Expertise During COVID-19"

Key results:

In 2020, Nature and several other scientific publications chose to endorse a presidential candidate. This article describes an experiment documenting effects of this endorsement. The effects are not positive and offer important insights for Nature and science-organizations in general.

The experiment was conducted on >4000 people in the US. The treatment group saw Nature's presidential endorsement with a note that Nature is one of the most cited scientific journals in the world. The control group saw Nature's announcement about a formatting change.

Findings:

1. The endorsement did not have a significant positive influence on left-leaning persons' trust in science or any other science related attitudes that the researchers measured. Specifically, "there is little evidence that the endorsement changed subjects' opinions about the two presidential candidates with respect to the issues that the endorsement piece highlights."
2. The endorsement led right-leaning people to be far less trusting of Nature and of science in general. Specifically: "Treated Trump and Biden supporters become two to four times more polarized than

control subjects on these stated measure of trust in Nature. The effect is greater for subjects who did not expect Nature to make an endorsement ex ante. The endorsement also significantly dampened Trump supporters' demand for COVID-related information provided by Nature."

3. In other words, this effect made these people far less trusting of other information coming from Nature. Specifically, "When prompted to acquire information about emerging COVID-19 variants and vaccine efficacy, treated Trump supporters were 38% less likely than control Trump supporters to request stories from Nature's website, indicating the decrease in stated trust has behavioural consequences. The endorsement also significantly reduced trust in scientists in general among Trump supporters, creating a "reputational externality" on the entire scientific community."

This is one of the two or three most important science communication articles that I have read in the last five years.

Validity: The experimental design is simple and straightforward. Analyses come from a preregistered design and are described in sufficient detail to permit accurate interpretation. The authors do not overclaim when interpreting their analysis.

Originality and significance: This is a very important paper. It shows that Nature's decision to endorse a presidential candidate likely produced very different consequences than it anticipated. Given that other scientific organizations have been, or may be tempted, to make the same choice, this article provides strong evidence for rethinking such actions. The findings should be of very broad interest – they can speak to any field of science.

Data & methodology: The data come from a simple but rigorous experimental design. A pre-registered analysis plan allows the authors to adapt to variations in what types of people do, or do not, participate in the experiment. The experiment, embedded in a survey, follows all established best practice in the field of survey-based experimentation. The data and design are described in sufficient detail to allow readers to interpret results accurately. The design is described in sufficient detail to allow an exact replication.

Preregistration: The authors follow the preregistration plan that is attached to this article. The authors do not explicitly call out all analyses possible with their data, they accurately characterize these findings and make them available in supplementary materials.

Appropriate use of statistics and treatment of uncertainties: Given the binary nature of the experimental treatment, the statistics used are sufficient. In the preanalysis plan, the authors cite a set of covariates that can influence effect sizes. They introduce these covariates into the analyses in ways that are consistent with the description in their analysis plan.

Custom code: n/a

Conclusions: This is one of the more important science communication papers that I have read in the last few years. It reveals unintended, and largely negative, consequence of mixing science with politics. Not only does Nature's decision to endorse a presidential candidate not have the effects that Nature's leadership may have intended, the study also shows that Nature's presidential endorsement had negative consequences for other kinds of trust in Nature's other content and created a negative

externality for the entire scientific community by reducing trust in science more generally considered.

Suggested improvements: My main question about this article is why it is not appearing in Nature. I am a big fan of Nature: Human Behavior, but this finding should be in Nature because (1) the importance of the findings applies across the sciences and (2) Nature's leadership should use this information to explain to the scientific community, whose reputation is at stake in such matters, how it will make future editorial decisions

References: This article references previous literature appropriately and in impressive detail.

Clarity and context: The argument and associated documentation is presented clearly and effectively.

Author Rebuttal to Initial comments

Dear Reviewer 1,

Thank you for your thoughtful, encouraging, and constructive review.

I have made my best attempts to address all your concerns. The results are reflected in the revised manuscript. Please find my response to your each comment below/

1, The conclusion of the paper—“(t)his study shows that electoral endorsements by scientific organizations reduce public trust in the endorser, particularly among supporters of the out-party candidate”—is overstated in my view. The study clearly shows that this CAN happen and that the effect can potentially be quite large, but it does not show that it DID happen in the 2020 U.S. Presidential election (only in a lab study conducted after the fact), or that it WILL happen in other instances in the future. In other words, I encourage the author to tone down the conclusion and make clear what was actually proven (versus the broader set of concerns that these important findings pose).

This is a very valid critique. Thanks again for pointing this out.

I have changed the sentence to (underscores added for the response letter):

“These results show political endorsement by scientific journals can undermine and polarize public confidence in the endorsing journals and in the scientific community.”

I have also added the following paragraph discussing the limitations of my finding to the discussion section, in which I addressed this shortcoming:

“This study also has several limitations and qualifications worth pointing out. First, the experiment demonstrates that the Nature endorsement decision can cause large negative reputation effects, but it does not show the endorsement did have such an impact or that other similar statements from scientific journals must produce such outcomes. In addition, the finding of sizable negative reputation externality on the scientific community as a whole may not be very widely generalizable beyond well-known top journals like Nature, which may be seen as distinctly exemplar of the scientific community because of its reputation (Herr 1987; Bless and Schwarz 2010). Finally, with a one-shot experiment, it cannot be proved that the effects found are long-lasting.”

2, A related point is that the paper currently does not speak to either limitations of the study, or to priorities for future research. Given the importance of the findings, I feel strongly both of these features must be included.

Again, a very valid critique.

To address this concern, I have also added the following paragraph discussing the limitations of my finding to the discussion section:

“This study also has several limitations and qualifications worth pointing out. First, the experiment demonstrates that the Nature endorsement decision can cause large negative reputation effects, but it does not show the endorsement did have such an impact or that other similar statements from scientific journals must produce such outcomes. In addition, the finding of sizable negative reputation externality on the scientific community as a whole may not be very widely generalizable beyond well-known top journals like Nature, which may be seen as distinctly exemplar of the scientific community because of its reputation (Herr 1987; Bless and Schwarz 2010). Finally, with a one-shot experiment, it cannot be proved that the effects found are long-lasting.”

“Future research should address some of the weaknesses of this study. Most importantly, techniques like follow-up surveys should be employed to determine, as soon as possible, whether the attitudinal and behavioral effects found in this study are long-lived. Further, future research should also examine if an explicit endorsement is necessary to produce these effects - would the impacts be different if the editorial statement makes the same criticism of the Trump administration without explicitly endorsing an alternative? Finally, it would also be of interest to explore empirically

whether these findings only apply to scientific journals or are they generalizable to other types of organizations.”

3, The title of the paper, and the body of the paper, refers to Nature as a “scientific organization.” It is more accurate to call Nature a “scientific journal” and to limit claims to scientific journals rather than scientific organizations. The boomerang effects identified may—indeed, is likely to—also occur when scientific organizations (e.g., AAAS) make political endorsements, but that has not yet been tested or proven.

Apologies for framing the results excessively broadly. To address this, I have replaced “organizations” with “journals” in many places. Most importantly, the title of the paper has been changed to: *“Political Endorsement by Scientific Journals and Trust in Scientific Expertise During COVID-19”*.

I kept the word “scientific organizations” in only one place: the first sentence of the introduction:

“Scientific organizations and publications have become increasingly involved in electoral politics.”

The idea here is I hope to *motivate* the research more broadly. However, when framing my results, the word “journals” is invariably used instead of “organizations” in the revised version. For example, the last sentence of the abstract now reads:

“These results show political endorsement by scientific journals can undermine and polarize public confidence in the endorsing journals and the scientific community.”

I would appreciate if you could let me know if this was an appropriate approach!

4, The paper addresses relevant theory only superficially in the introduction (although, oddly, it includes an extensive summary of the findings—which takes up approximately half of the introduction section), and not at all in the discussion. I suspect the superficial reference to theory in the introduction was in interest of moving readers quickly into the research itself—which is fine—but the paper will be stronger if some consideration of theory is incorporated into the discussion section.

I am sorry for the inadequate discussion of theories. I have added the following paragraph to the discussion section:

“It is instructive to interpret the results in the the light of science communication theories. Donner(2014) proposes conceptualizing advocacy in scientists' public

communications as a continuum, where higher levels of advocacy are characterized by more normative judgements and greater influence of the scientist's worldview. He also theorizes that greater degrees of advocacy leads to greater "professional risks" like "alienating those ... audience with opposing political views.". This phenomenon that messages can damage the the persuasiveness of the communicators is referred to as "source derogation" in communication and psychology research (Zuwerink and Cameron 2003). The type of political endorsement studied here, which advocates voting as a tactic to bring about better policy actions, is close to the highest level of advocacy Donner(2014) considered, which is characterized by being motivated by questions like "should certain tactics be employed to bring about specific action? e.g., participate in protest against fossil fuel development or pipeline)". This study's findings thus strongly support Donner's hypothesis that high level of advocacy leads to high risks to the communicator's perceived credibility; the difference between these findings and that of Kotcher et al. (2017), that climate-related policy advocacy has limited or no effect on the scientist's credibility, also fits the theory, since supporting specific policies ("actions") corresponds to a lower level of advocacy than advocating political "tactics" (voting in this case) to bring about these actions on Donner's continuum. This study can thus be seen as complementary to Kotcher et al. (2017) in this light. In addition, this study's finding showing effect heterogeneity by the audience's political predisposition is consistent with research on motivated reasoning, i.e., the tendency for agents to systematically reject information that contradicts their own deeply-held beliefs (Greenwald and Ronis 1978; Leeper and Slothuus 2014)."

I would appreciate it if you could let me know if there were any other theories I should discuss!

5, The author speculates that the impacts of learning about the endorsement are likely to be long-lived, but frankly, I found the author's argument to be hard to follow; it should be clarified. More importantly, it should be clearly acknowledged that the impacts may NOT be long-lived, and that additional research should be conducted, ASAP, to determine the duration of impacts.

Thanks for the callout. My statement is a stretch. I have deleted any statement that the effect may be long-lasting from the paper, and I have included remarks about this limitations and the need for future research in the new "limitation and suggestions for future research" paragraphs in the discussion section:

"This study also has several limitations and qualifications worth pointing out. First, the experiment demonstrates that the Nature endorsement decision can cause large negative reputation effects, but it does not show the endorsement did have such an impact or that other similar statements from scientific journals must produce such

outcomes. In addition, the finding of sizable negative reputation externality on the scientific community as a whole may not be very widely generalizable beyond well-known top journals like Nature, which may be seen as distinctly exemplar of the scientific community because of its reputation (Herr 1987; Bless and Schwarz 2010). Finally, with a one-shot experiment, it cannot be proved that the effects found are long-lasting.

“Future research should address some of the weakness of this study. Most importantly, techniques like follow-up surveys should be employed to determine whether the attitudinal and behavioral effects found in this study are long-lived as soon as possible. Further, future research should also determine if an explicit endorsement is necessary to produce these effects - would the impacts be different if the editorial statement makes the same criticism of the Trump administration without explicitly endorsing an alternative? Finally, it would also be of interest to explore empirically whether these findings only apply to scientific journals or are they generalizable to other types of organizations.”

6, While the research was pre-registered, no hypotheses were stated in the pre-registration materials, nor in the paper itself. It isn't clear to me how readers should interpret pre-registered research questions (rather than pre-registered hypotheses). I would like to see the author address that issue in the discussion.

My apologies - my pre-registration materials are organized in a messy way and a bit confusing. The materials do include hypotheses. The preregistration materials include four pdf files:

- 1, AEA Preregistration.pdf
- 2, AEA Pre-Analysis Plan.pdf
- 3, OSF Preregistration.pdf
- 4, OSF Pre-Analysis Plan.pdf

The hypotheses are outlined in files 2, 3, and 4, but not in file 1. Again, I am sorry for the confusion.

7, Kotcher et al (2017) tested the impact of advocacy statements on trust in the communicating scientist and in the scientific community more broadly; they found limited to no impact. That research is sufficiently relevant to warrant consideration in the current paper. [Reference: Kotcher et al. (2017) Does engagement in advocacy hurt the credibility of scientists? Results from a randomized national survey experiment. Environmental Communication.]

Thank you very much for pointing this out. I was not aware of this highly relevant paper. It would be a mistake to omit it.

I have added the following sentence to the introduction section:

“Also closely related is an experimental study by Kotcher et al. (2017), which finds climate-related policy, instead of political, advocacy has limited or no effect on the perceived credibility of the communicating scientists and the scientific community. “

I also discuss this paper further in the newly added theory paragraph in the discussion section:

“the difference between these findings and that of Kotcher et al. (2017), that climate-related policy advocacy has limited or no effect on the scientist's credibility, also fits the theory, since supporting specific policies (“actions”) corresponds to a lower level of advocacy than advocating political “tactics” (voting in this case) to bring about these actions on Donner's continuum. This study can thus be seen as complementary to Kotcher et al. (2017) in this light.”

8, Table 1 could be strengthened in the following ways:

- **Drop the N column (as it is largely redundant with the % column)**
- **Add a column to show % from the current census data (to compare the sample to the overall population)**
- **I suggest limiting the numbers to one place after the decimal (e.g., 46.5%)**

This is a very useful suggestion. I have implemented the change. Thank you.

9, There is inconsistency in how one of the “trust in Nature” variables is referred to in the paper. In some places it is referred to as “informed” and in other places “competent.”

Thanks for the call-out. I have replace all “competence” with “informedness”.

10, On page 5, the verbs in the 2nd paragraph are in the present tense (e.g., are) rather than in the past tense (were). Given that the actions occurred in the past, I think referring to them as past is appropriate.

and

11, There is a misspelling in the first sentence of the 2nd paragraph of the discussion: “results”.

Thanks for noticing and pointing out these grammatical and spelling errors. I have fixed them in the revised manuscript.

I am grateful for all these comments and suggestions, which have strengthened the paper greatly.

Best regards,
The author

Dear Reviewer 2,

Thank you for your thoughtful, encouraging, and constructive review.

I appreciate your suggestion that this paper should appear in *Nature*. (“My main question about this article is why it is not appearing in Nature. I am a big fan of Nature: Human Behavior, but this finding should be in Nature because (1) the importance of the findings applies across the sciences and (2) Nature’s leadership should use this information to explain to the scientific community, whose reputation is at stake in such matters, how it will make future editorial decisions”)

I think *Nature Human Behaviour* is an excellent journal, which has a great record of publishing high-quality and high-impact interdisciplinary social science studies and I would be honoured to publish this paper in the journal.

Thank you very much for reviewing the paper and for your very positive assessments, which are very heartening to read!

Best regards,
The author

Decision Letter, first revision:

21st November 2022

Dear Dr. Zhang,

Thank you for submitting your revised manuscript "Political Endorsement by Scientific Journals and Trust in Scientific Expertise During COVID-19" (NATHUMBEHAV-22061390A). It has now been seen by the original referees and their comments are below. As you can see, the reviewers find that the paper has improved in revision. We will therefore be happy in principle to publish it in Nature Human Behaviour, pending minor revisions to satisfy the referees' final requests and to comply with our editorial and formatting guidelines.

We are now performing detailed checks on your paper and will send you a checklist detailing our editorial and formatting requirements within a week. Please do not upload the final materials and make any revisions until you receive this additional information from us.

Sincerely,

Samantha Antusch

Samantha Antusch, PhD
Senior Editor
Nature Human Behaviour

Reviewer #2 (Remarks to the Author):

Thank you for sending the revised version of the article. I agree with the queries raised by the first reviewer and I am grateful for the author's responses. I think that the paper is much stronger as a result.

I found two typos: "assigns" in the abstract and "especial" on page 5.

This article makes an important contribution to several literatures. The new emphasis on limitations not only increases the truth-value of a number of important claims, but also provides a clearer guide to future research that can help scientific journals and organizations better serve society while maintaining their credibility. I am glad to see methodologically sophisticated people doing this type of work.

Final Decision Letter:

Dear Dr Zhang,

We are pleased to inform you that your Article "Political Endorsement by Nature and Trust in Scientific Expertise During COVID-19", has now been accepted for publication in *Nature Human Behaviour*.

Please note that *Nature Human Behaviour* is a Transformative Journal (TJ). Authors whose manuscript was submitted on or after January 1st, 2021, may publish their research with us through the traditional subscription access route or make their paper immediately open access through payment of an article-processing charge (APC). Authors will not be required to make a final decision about access to their article until it has been accepted. IMPORTANT NOTE: Articles submitted before January 1st, 2021, are not eligible for Open Access publication. Find out more about Transformative Journals

With best regards,

Samantha Antusch

Samantha Antusch, PhD
Senior Editor
Nature Human Behaviour